# Tailoring amorphous boron nitride for high-performance two-dimensional electronics

Cindy Y. Chen [1], Zheng Sun[2], Riccardo Torsi[1], Ke Wang[3], Jessica Kachian[4], Bangzhi Liu[3], Gilbert B. Rayner Jr[5], Zhihong Chen [2], Joerg Appenzeller [2], Yu-Chuan Lin [6] ✉ & Joshua A. Robinson[1,3,7] ✉

Two-dimensional (2D) materials have garnered significant attention in recent years due to their atomically thin structure and unique electronic and optoelectronic properties. To harness their full potential for applications in next-generation electronics and photonics, precise control over the dielectric environment surrounding the 2D material is critical. The lack of nucleation sites on 2D surfaces to form thin, uniform dielectric layers often leads to interfacial defects that degrade the device performance, posing a major roadblock in the realization of 2D-based devices. Here, we demonstrate a wafer-scale, low-temperature process (<250 °C) using atomic layer deposition (ALD) for the synthesis of uniform, conformal amorphous boron nitride (aBN) thin films. ALD deposition temperatures between 125 and 250 °C result in stoichiometric films with high oxidative stability, yielding a dielectric strength of 8.2 MV/cm. Utilizing a seed-free ALD approach, we form uniform aBN dielectric layers on 2D surfaces and fabricate multiple quantum well structures of aBN/MoS$_2$ and aBN-encapsulated double-gated monolayer (ML) MoS$_2$ field-effect transistors to evaluate the impact of aBN dielectric environment on MoS$_2$ optoelectronic and electronic properties. Our work in scalable aBN dielectric integration paves a way towards realizing the theoretical performance of 2D materials for next-generation electronics.

Two-dimensional (2D) materials such as graphene and transition metal dichalcogenides (TMDs) are considered as promising, alternative materials for augmenting conventional Si-based technology due to their atomically thin structure and tunable electronic and optical properties[1-3]. Although promising for the continued scaling of transistors, 2D materials can also present several challenges that prevent the realization of their theoretical performance. First, 2D materials are highly susceptible to ambient instability, as the presence of surface

defects can more profoundly impact the 2D material's intrinsic electronic properties compared to that of bulk materials[4-6]. Additionally, the integration of a suitable dielectric environment in 2D material-based transistors is critical for enhancing the carrier transport properties of 2D semiconductors[7-10]. However, the lack of out-of-plane bonding in 2D van der Waals (vdW) surfaces adds considerable difficulty for the integration of gate dielectrics on 2D materials, as the chemical inertness leads to lower growth rates and non-uniform

[1]Department of Materials Science and Engineering, The Pennsylvania State University, University Park, PA 16802, USA. [2]School of Electrical and Computer Engineering and Birck Nanotechnology Center, Purdue University, West Lafayette, IN 47907, USA. [3]Materials Research Institute, The Pennsylvania State University, University Park, PA 16802, USA. [4]Intel Corporation, 2200 Mission College Blvd, Santa Clara, CA 95054, USA. [5]The Kurt J. Lesker Company, 1925 PA-51, Jefferson Hills, PA 15025, USA. [6]Department of Materials Science and Engineering, National Yang Ming Chiao Tung University, Hsinchu City 300, Taiwan. [7]Two-Dimensional Crystal Consortium, The Pennsylvania State University, University Park, PA 16802, USA. ✉e-mail: ycl194@nycu.edu.tw; jar403@psu.edu

growth of dielectric layers on 2D materials[11,12]. To improve 2D device performance, it is critical to develop an encapsulation process for material passivation and the integration of dielectrics, with specific considerations of back-end-of-line (BEOL) temperature requirement, scalability, reliability, and impact of dielectric environment on 2D electronic properties.

Strategies for dielectric integration include atomic layer deposition (ALD) of 3D amorphous oxides, or transfer of hexagonal boron nitride (hBN) grown via chemical vapor deposition (CVD). ALD of high-$\kappa$ dielectrics such as $Al_2O_3$ and $HfO_2$ on TMDs can be carried out on a wafer-scale, at temperature <300 °C, and with layer-by-layer control. However, water-based precursors are commonly used to synthesize ALD oxides, resulting in the continued possibility of in situ oxidation and degradation of the TMDs[13]. As ALD oxides are 3D in structure, dangling bonds and charged impurities can form at the interface with the 2D semiconductor channel, resulting in the decrease of carrier mobility due to additional charge scattering. In contrast, hBN layers are atomically smooth and can form a clean vdW interface with the 2D semiconducting channel, which can substantially reduce charge carrier scattering due to surface roughness and charged impurities[14]. hBN encapsulation also leads to reduced remote phonon scattering since the high energy surface optical phonon modes of hBN do not couple to the low energy modes in 2D semiconductors[15]. While hBN is considered a more suitable dielectric material for 2D encapsulation, tradeoffs are present in scalability and thermal requirement, as the CVD of hBN is carried out at high temperatures (>1000 °C) and on metal templates, and often accompanied by mechanical exfoliation and transfer processes onto the TMD[16]. In comparison, amorphous BN (aBN) can be readily synthesized at low temperature, making it a promising solution for overcoming the stringent processing requirements of hBN. Nevertheless, the scalable synthesis and impact of aBN on 2D semiconductors still requires further investigation.

Here, we present the ALD of ultrathin (2–20 nm) aBN as a scalable, non-water-based, low-temperature process for dielectric integration with 2D semiconductors. We evaluate the impact of ALD processing parameters on the resulting morphology, chemical composition, and structural properties of aBN and demonstrate uniform integration of ultrathin aBN on $MoS_2$ to enable demonstration of $aBN/MoS_2$ quantum wells. Finally, we fabricate and characterize ALD aBN-encapsulated double-gated monolayer (ML) $MoS_2$ transistors with various channel lengths and gate dielectric stacks to study key performance metrics across a large number of devices.

## Results and discussion
### ALD synthesis of aBN
We demonstrate the wafer-scale, uniform, and conformal deposition of BN on planar Si, structured wafers, and 2D material surfaces using ALD. BN thin films are deposited using ALD with sequential flows of $BCl_3$ and $NH_3$, which react under the following overall (net) ligand exchange reaction[17–19]: $BCl_3$ (g) + $NH_3$ (g) → BN (s) + 3HCl (g). A process schematic demonstrating the BN ALD reaction cycle is presented in Fig. 1a. Unless otherwise specified, BN ALD process and substrate preclean conditions presented in this work are those described under the "Methods" section. Details of the ALD process optimization are described in Supplementary Fig. 1, where we show that a soak time >2 s between the precursor pulse and purge steps and between the reactant

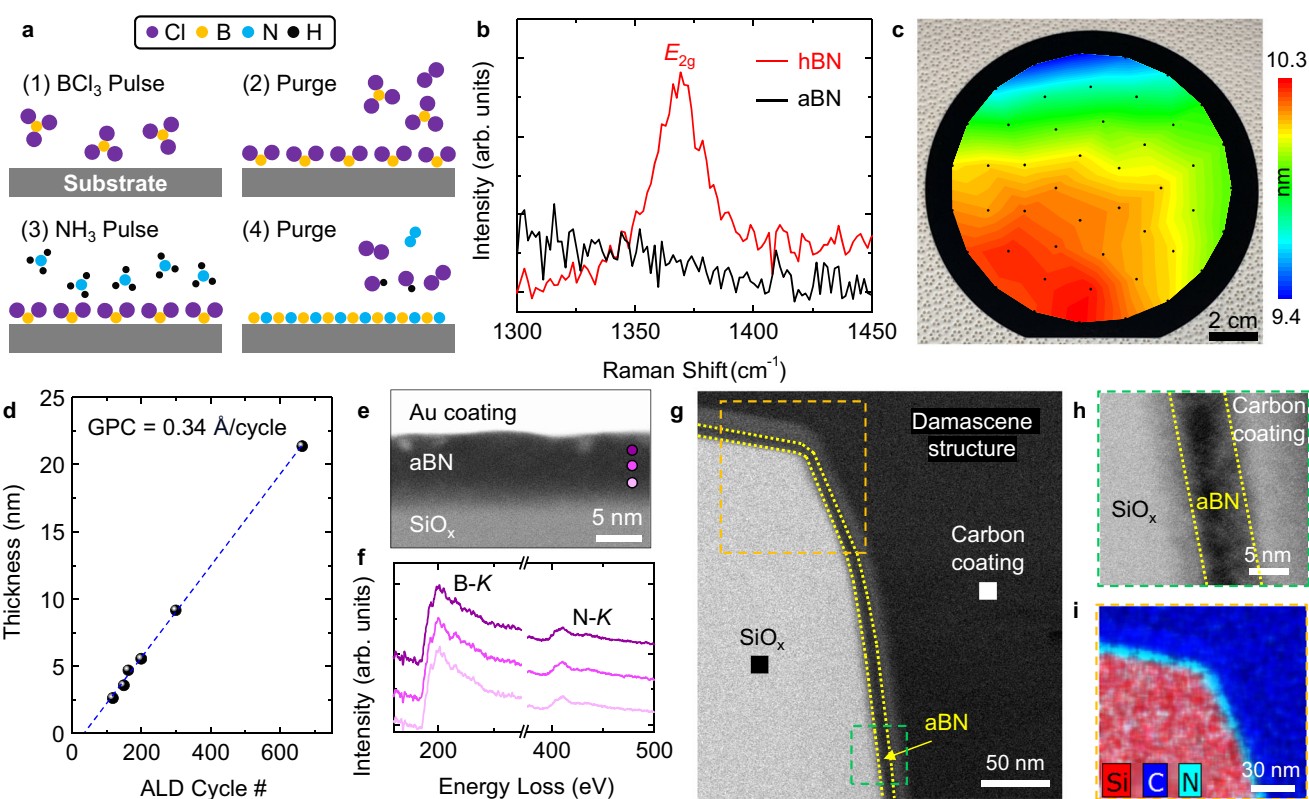

**Fig. 1 | Wafer-scale atomic layer deposition of amorphous boron nitride.**
**a** Schematic illustration of the ALD process for BN thin films. **b** Raman spectra of aBN (black) and hBN (red). The in-plane $E_{2g}$ vibrational mode of hBN is observed at 1369 cm⁻¹, which is absent in aBN due to the lack of crystalline order. **c** Camera image and overlaid map of film thickness measured by spectroscopic ellipsometry of as-deposited aBN (300 ALD cycles) on a 150 mm Si wafer. **d** Thickness of aBN deposited at 250 °C as a function of ALD cycle number. A linear fit (blue) is used to obtain a growth rate of 0.34 Å/cycle after the initial nucleation delay.

**e** HAADF-STEM cross-sectional image of Au-capped aBN deposited on $SiO_x$ at 250 °C for 300 ALD cycles. **f** EELS characteristics of B-$K$ and N-$K$ from the selected points in (**e**). **g** HAADF-STEM cross-sectional image of aBN deposited at 250 °C for 300 ALD cycles on a damascene structure made of $SiO_x$. The aBN is highlighted with yellow dotted lines. **h** Cross-sectional HAADF-STEM image in the green selected area in (**g**). **i** EDS mapping of Si, C, and N in the orange selected area in (**g**).

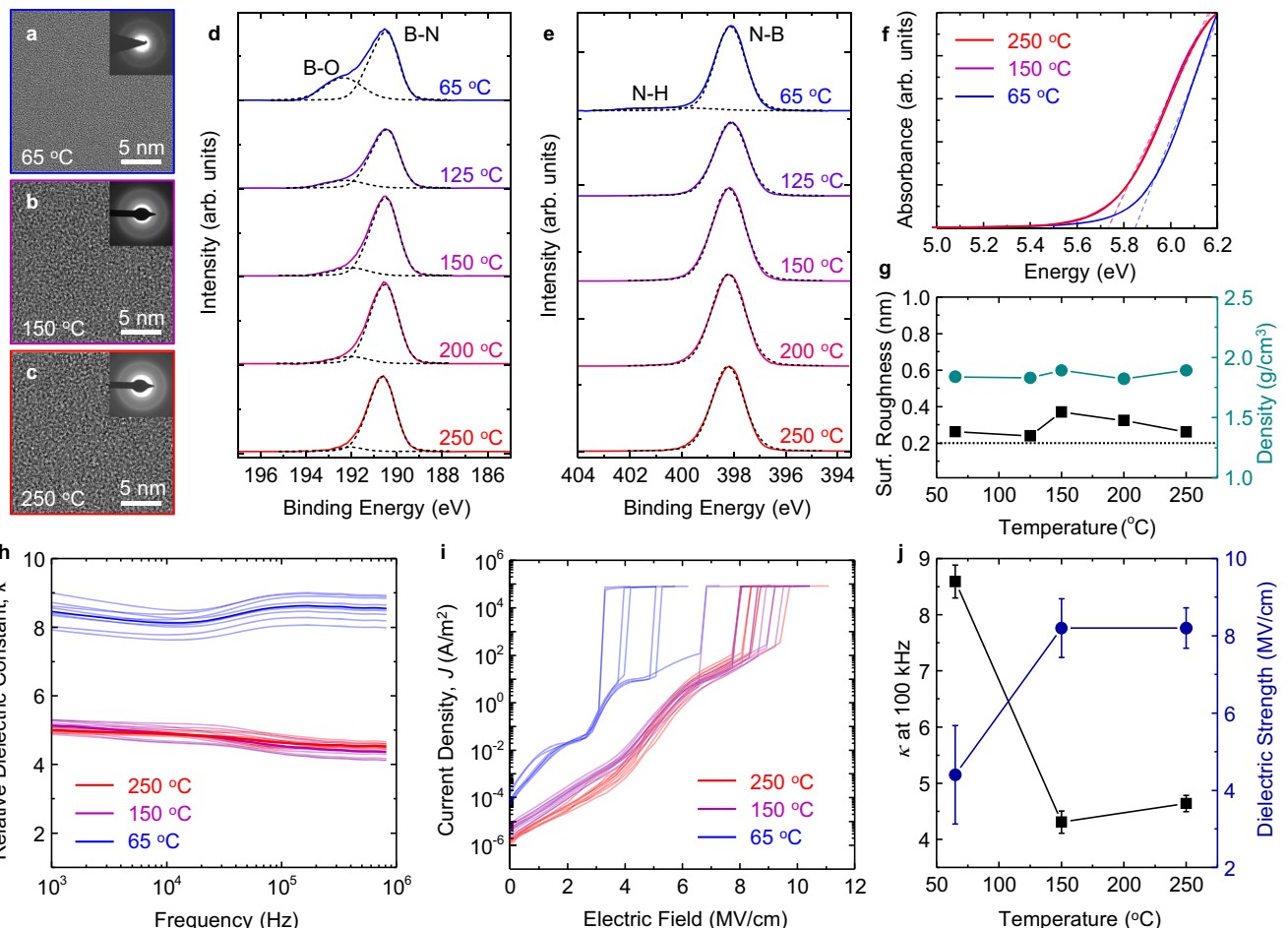

**Fig. 2 | Deposition temperature-dependent properties of amorphous boron nitride.** Plan-view HRTEM image and the corresponding SAED pattern (inset) of aBN deposited for 300 cycles onto amorphous $SiO_2$ TEM grids at **a** 65, **b** 150, and **c** 250 °C. **d** B 1*s* and **e** N 1*s* core level XP spectra of aBN deposited for 300 cycles on Si from 65 to 250 °C. **f** Tauc plots of aBN with linear fits (dashed lines) for the estimation of aBN optical bandgaps, which are 5.7 eV at 250 and 150 °C, and 5.8 eV at 65 °C. **g** Surface roughness (black) and density (green) of aBN deposited via 300

ALD cycles on Si at 65–250 °C. Dotted line shows bare Si substrate roughness for reference. **h** Relative dielectric constant vs. frequency for aBN deposited from 65 to 250 °C for 300 cycles. **i** Current density vs. electric field for dielectric strength of aBN deposited at different temperatures (same deposition conditions as for (**h**)). **j** Relative dielectric constant at 100 kHz (black) and dielectric strength (blue) of aBN deposited at different temperatures (same deposition conditions as for (**h**)).

pulse and purge steps significantly improves adsorbate surface coverage after each half-cycle and yields higher growth rate and wafer-scale thickness uniformity. Due to the surface controlled, self-limiting growth mechanism of ALD, wafer-scale thin films can be deposited with highly controllable thicknesses and low temperature ranges compatible with BEOL process temperatures[20]. The lack of thermal energy in ALD generally inhibits diffusion of precursor molecules[21], indicating that the BN films will be amorphous or nanocrystalline. For the following structural characterization of ALD-deposited BN, we prepare BN samples deposited at 250 °C for 300 cycles. The structure of ALD-deposited BN on Si is investigated by Raman spectroscopy using a 532 nm excitation laser and measuring the phonon lines between 1300 and 1450 cm$^{-1}$. For structural comparison, we also collect Raman spectra of CVD-grown hBN[22], which exhibits the characteristic Raman-active in-plane transverse $E_{2g}$ mode at 1369 cm$^{-1}$ (Fig. 1b). The $E_{2g}$ mode is not observed for the ALD-deposited BN film, indicating a lack of crystalline order and thus confirms the amorphous nature of the thin film. To assess the scalability of the ALD process, the thickness profile of aBN deposited at 250 °C on a 150 mm Si wafer is mapped with spectroscopic ellipsometry (Fig. 1c). At 300 ALD cycles, the average thickness is 9.7 ± 0.2 nm across the wafer. We further characterize the ALD growth characteristics of aBN on Si by evaluating the film thickness at different ALD cycle numbers, as shown in Fig. 1d.

After an initial nucleation delay, a linear dependence of the film thickness with increasing ALD cycle is observed, which is consistent with the self-saturating nature of surface reactions during ALD half cycles. The growth rate, defined as the growth per cycle (GPC), is 0.34 Å/cycle based on the slope of the linearly fitted region. This is within the range of GPC values (0.32–0.42 Å/cycle) reported in previous studies on thermal ALD reactions with $BCl_3$ and $NH_3$ on Si substrates[17,23].

The ALD process also enables the conformal coverage of aBN on structured surfaces, a key technological requirement in ultra-scaled applications. Cross-sectional, high-angle annular dark field scanning transmission electron microscopy (HAADF-STEM) is performed to assess the in-plane uniformity of aBN deposited at 250 °C for 300 cycles on $SiO_x$. To provide sufficient contrast for layer distinction, 30 nm Au is thermally evaporated onto the aBN layer. HAADF-STEM image (Fig. 1e) shows a clear interface between aBN and the $SiO_x$ substrate and continuity with no presence of pinholes. The presence of B and N is further confirmed via electron energy loss spectroscopy (EELS) analysis at different points across the film thickness (Fig. 1f). To assess the effectiveness of the ALD process for conformal aBN deposition, we carry out aBN deposition at 250 °C for 300 cycles on $SiO_x$ substrates patterned with 10 by 10 µm² trenches 500 nm in depth. The cross-sectional STEM image in Fig. 1g and Supplementary Fig. 2

indicates uniform thickness over the top and sidewall of SiO$_x$, confirming the conformality of aBN. It should be noted that the thin layer of carbon coating on top of aBN is deposited via electron beam evaporation, where the higher density of the carbon layer leads to higher intensity and better contrast for assessment of aBN thickness uniformity on SiO$_x$. As a result, the aBN layer can be clearly observed under the higher-magnification view of the sidewall (Fig. 1h). Energy-dispersive X-ray spectroscopy (EDS) mapping showing N signals conformally confined by Si and C signals at the trench edge further validates the uniform deposition of aBN (Fig. 1i). B signals are not shown due to detection limits of EDS for lighter elements.

**Deposition temperature-dependent properties of aBN**

The growth rate of aBN exhibits dependence on the ALD deposition temperature. In ALD processes, the deposition temperature plays an important role in surface chemistry and reaction kinetics, which in turn dictate the film properties[21]. In this section, we elucidate the impact of deposition temperature, measured from the substrate heater stage, on film properties of aBN deposited from 65 to 250 °C for 300 cycles. The GPC of aBN on a Si wafer increases from 0.31 to 0.59 Å/cycle as the deposition temperature decreases from 250 to 65 °C (Supplementary Fig. 3a). The higher GPC at low deposition temperatures suggests a deviation from the ALD self-limiting growth behavior, which is likely due to the low-temperature physisorption of multiple monolayers of precursor at the substrate surface[24,25].

Varying the ALD deposition temperature from 65 to 250 °C does not result in changes in BN crystallinity. We evaluate the film crystallinity by depositing aBN onto amorphous SiO$_2$ TEM grids for high-resolution transmission electron microscopy (HRTEM) analysis. HRTEM images of aBN deposited from 65 to 250 °C do not exhibit long-range order present in hBN (Fig. 2a–c). Additionally, diffuse diffraction in selective-area electron diffraction (SAED) patterns further demonstrate the lack of preferred crystallographic orientation (inset, Fig. 2a–c).

The ALD deposition temperature plays an essential role in oxygen content in aBN films on Si. This is evident using x-ray photoelectron spectroscopy (XPS), where high-resolution spectra in the B 1$s$ region are fit with two components: B-N at 190.5 eV and B-O at 192.4 eV (Fig. 2d)[26]. No B-Cl component, expected at 193.5 eV in the B 1$s$ region, is observed within the detection limit of XPS[27]. The B-O component peak intensity increases as deposition temperature decreases, indicating an increase in the O incorporation into the aBN film. The total O incorporation into the film as quantified from the B-O component increases from 2.1% at 250 °C to 11.4% at 65 °C (Supplementary Fig. 3c). The O component is not observed in N 1$s$ spectra, where only the N-B component is identified at 398.2 eV, and a low concentration of N-H component is identified at 400.2 eV for aBN deposited at 65 °C (Fig. 2e). The B/N ratio quantified from B and N 1$s$ core level spectra is 0.9 from 125 to 250 °C, and 1.1 at 65 °C (Supplementary Fig. 3b). The N-deficiency shown in the 65 °C film, combined with the higher detected O at%, suggests that oxidation is favorable when aBN is N-deficient. This is consistent with oxidation of defects in hBN, where N vacancies are decorated with O (O$_N$)[28]. Compared to other defect systems such as O at B sites (O$_B$) or C substitutional defects (C$_B$ or C$_N$), O$_N$ has the lowest formation energy of 2.20 eV[29]. The N-deficiency shown in the 65 °C film, combined with the higher detected O at%, is also consistent with incomplete ligand exchange at low deposition temperatures, leading to oxidation of unreacted B-Cl bonds upon air exposure and noting all XPS measurements are ex situ. This model is further supported by the presence of the N-H component of the N 1$s$ peak and the absence of the B-Cl component within the B 1$s$ peak in the XPS of aBN deposited at 65 °C. Based on these results, it is expected that aBN films formed at higher deposition temperatures are stoichiometric and, therefore, more resistant to oxidation. To evaluate this model, which posits that detected aBN oxidation occurs ex situ, we

synthesize an aBN stack consisting of an initial deposition at 65 °C, followed by deposition at 200 °C without exposure to ambient. We note that, for aBN deposited at temperatures <125 °C, the Si concentration approaches the detection limit of the XPS tool (Supplementary Fig. 3c). This corresponds to an aBN thickness limit of 13 nm, above which the Si substrate is not detectable by XPS. The hybrid aBN film stack is 6.6 nm and yields an oxidation level that is notably reduced compared to that estimated for the hybrid aBN film stack using oxidation levels of the pure 65 °C- (i.e., aBN resulting from deposition at 65 °C only) and pure 200 °C- (i.e., aBN resulting from deposition at 200 °C only) aBN films and accounting for the 65 °C- and 200 °C-aBN ALD thickness contributions to the hybrid aBN film stack (Supplementary Fig. 4). The estimated (versus measured) O at% in the hybrid aBN film stack is >2× higher. Note that if oxidation occurs in situ, then the hybrid aBN film stack would yield equal measured and estimated oxidation levels. Consequently, the data supports that observed aBN oxidation occurs ex situ. Further, the data set cannot be used to evaluate the ability of the 200 °C aBN component film of the hybrid aBN film stack to hermetically seal the underlying 65 °C aBN component film. Completing the aforementioned evaluation requires that 200 °C aBN ALD (second ALD step of the in situ hybrid aBN film stack deposition) negligibly transforms the 65 °C aBN component film (deposited in the first ALD step of the in situ hybrid aBN film stack deposition). If this requirement is met, a good hermetic seal would be supported by XPS detection of B-Cl in the underlying 65 °C aBN component film (via a B-Cl binding energy component in the B 1$s$ envelope), and given component film thicknesses, a bad hermetic seal would be supported by equal measured and estimated oxidation levels for the hybrid aBN film stack. That we observe neither the former nor the latter by XPS of the hybrid aBN film stack suggests that 200 °C aBN ALD (second ALD step of the in situ hybrid aBN film stack deposition) significantly transforms the 65 °C aBN component film (deposited in the first ALD step of the in situ hybrid aBN film stack deposition). This is consistent with the 200 °C aBN ALD step of the in situ hybrid aBN film stack deposition driving increased extent of reaction within the underlying 65 °C aBN component film through elevated temperature allowing unreacted B-Cl and N-H bonds within the 65 °C aBN component film to react with each other and/or with incoming precursors (that can access sites within the 65 °C aBN component film due its thickness at or below minimum continuous thickness or courtesy of diffusion between domains). This is also consistent with similar measured O at% values of 3% (Supplementary Fig. 4) and 4% (Supplementary Fig. 3c) for the hybrid aBN film stack and the pure 200 °C aBN film, respectively. Overall, the data in this section support that aBN film deposition solely at higher temperatures or, pending component film thicknesses, in situ hybrid aBN film stack deposition comprised of low-then high-temperature aBN ALD steps yields aBN films or film stacks more resistant to post-growth oxidation, respectively. Therefore, to achieve stoichiometric aBN films or film stacks and maintain high oxidative and hydrolytic stability of aBN, aBN ALD should be performed solely at a deposition temperature within the 125–250 °C window or via an in situ hybrid aBN film stack deposition process comprised of relatively thin aBN deposition at 65 °C followed by relatively thick aBN deposition at a temperature within the 125–250 °C window[30]. As subsequent sections will show, the latter approach can be used to grow thin aBN films on TMD surfaces without impact to aBN or TMD quality.

aBN films deposited via 300 ALD cycles at 65–250 °C on transparent quartz substrates for absorption measurements exhibit a 5.7–5.8 eV bandgap (Fig. 2f). This is comparable to crystalline hBN with bandgaps ranging from 5.6 to 6.0 eV, where the variation is strongly dependent on sample quality and the employed synthesis method[31–35]. Furthermore, we find that the ALD process yields atomically smooth aBN on Si with root-mean-square surface roughness ($R_q$) varying from 0.24 to 0.37 nm across the 65–250 °C window (Fig. 2g, black) with no

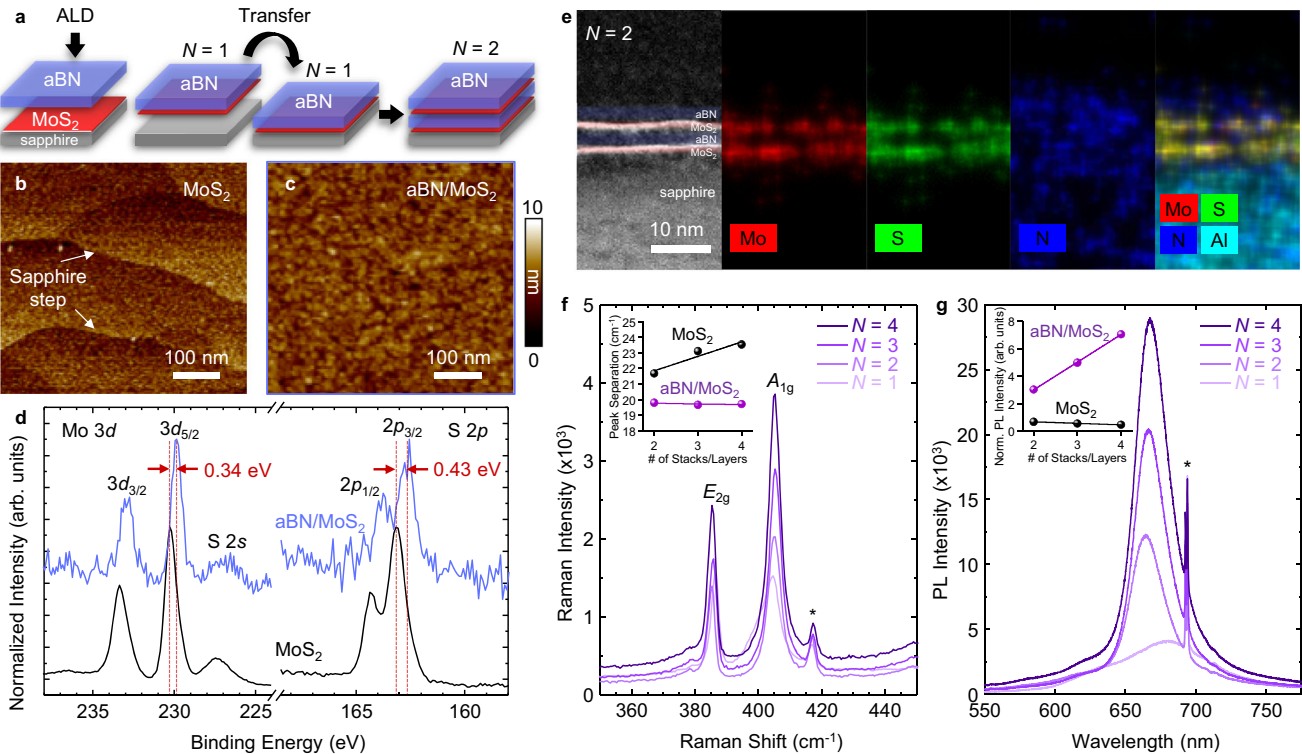

**Fig. 3 | Integration of aBN with MoS₂ for evaluation of optoelectronic properties. a** Schematic diagram of the synthesis of aBN/MoS₂ superlattices. AFM image of **b** as-grown monolayer MoS₂ and **c** aBN deposited on MoS₂. **d** Mo 3$d$ and S 2$p$ XP spectra of as-grown MoS₂ (black) and after aBN deposition (blue) with the same process conditions as (**c**). **e** Cross-sectional HAADF-STEM image and corresponding EDS mapping of $N=2$ aBN/MoS₂ superlattices on α−Al₂O₃. **f** Raman spectra in the range of MoS₂ $E_{2g}$ and $A_{1g}$ vibrational peaks for $N=1$–4 stacks of aBN/MoS₂. **g** PL spectra of $N=1$–4 stacks of aBN/MoS₂ showing increasing intensity as $N$ increases. In (**f**, **g**), * denotes the signal from the sapphire substrate, and the insets present a comparative analysis between multilayer MoS₂ and aBN/MoS₂ stacks. The PL intensity for MoS₂ in the inset of (**g**) is adapted from ref. 45.

discernible deposition temperature dependence. The thin film density of aBN on Si, probed using x-ray reflectivity (XRR), does not exhibit any deposition temperature dependence, and varies from 1.8 to 1.9 g/cm³ from 65 to 250 °C (Fig. 2g, green). These densities are lower than the theoretical density of hBN at 2.1 g/cm³, which is likely due to a lower network connectivity of the amorphous phase compared to the layered crystalline phase.

The dielectric response of aBN films varies with ALD deposition temperature for aBN deposited via 300 cycles of ALD on p⁺⁺-Si at 65, 150, and 250 °C. The capacitance measurements of Pt/aBN/p⁺⁺-Si metal-insulator-semiconductor (MIS) stacks from 10³ to 10⁶ Hz demonstrate the extracted relative dielectric constant, $\kappa$, decreases by ~10% for aBN deposited at 250 and 150 °C, but exhibits a 2× increase at 65 °C (Fig. 2h), suggesting a different polarizability mechanism at low deposition temperatures. The $\kappa$ value at 100 kHz is 4.3, 4.6, and 8.6 for 250, 150, and 65 °C, respectively. Under an applied bias, we see the current density of aBN films increases steadily until dielectric breakdown to current saturation (Fig. 2i). The calculated dielectric strength is 4.4 MV/cm at 65 °C, and 8.2 MV/cm for 150 °C and 250 °C. The relative dielectric constant $\kappa$ at 100 kHz and dielectric strength at different aBN deposition temperatures are summarized in Fig. 2j. Only the 65 °C aBN film exhibits anomalous dielectric performance, with both a low dielectric breakdown strength and high $\kappa$ value. To date, aBN is reported to exhibit dielectric constants of 1.8–5.5[36–39]. The observed changes in dielectric properties may be due to several factors, including material crystallinity, density, and composition. Because ALD-aBN density and physical structure remain constant across the investigated temperature window, these may be ruled out as the dominating factor for dielectric response changes. However, the aBN does exhibit increased O incorporation with decreasing deposition

temperature. The higher $\kappa$ observed for 65 °C aBN in this study is markedly different from other reported $\kappa$ of aBN, which are from near stoichiometric aBN, and therefore may have minimized dipole contribution from polar B-O components. Based on these findings, we also expect that higher deposition temperatures, such as 300 °C, may further decrease the dielectric constant due to O concentration reduction in aBN. The O level can have a direct impact on the dielectric properties of aBN. $O_N$ defects, for instance, introduce defect levels near the conduction band, thereby enhancing the material's metallic characteristics[29,40]. This, in turn, can create conductive pathways in the band structure of aBN thin films, leading to the observed, and highly variable, low-field breakdown. ALD-aBN deposited at temperatures >150 °C features excellent structural and dielectric strength compared to other reported forms of BN thin films[22,38].

**Fabrication and characterization of aBN/MoS₂ quantum wells**

A modified, two-step, ALD process enables seed-free formation of ultrathin, continuous aBN dielectric layers on 2D material surfaces. This enables fabrication of aBN-based devices, including aBN/MoS₂ quantum well stacks (Fig. 3a)[41]. Utilizing the 65 °C aBN as the nucleating interfacial layer, we can subsequently deposit uniform, stoichiometric aBN at 250 °C on MoS₂. The optimal 65 °C ALD cycle number, based on AFM analysis of surface uniformity and film coalescence, is 40 (Fig. 3b, c and Supplementary Fig. 5). XPS analysis of aBN/MoS₂ in the Mo 3$d$ and S 2$p$ regions indicate a decrease in Mo and S intensities after aBN encapsulation, which is expected due to the attenuation of photoelectrons from the dielectric layer (Supplementary Fig. 6). Normalized XP spectra reveal a binding energy shift of −0.34 eV in Mo 3$d_{5/2}$ and −0.43 eV in S 2$p_{3/2}$ after aBN encapsulation (Fig. 3d). Additionally, no other defect peaks are detected, indicating the compatibility of the

ALD-aBN process in preserving the as-grown quality of monolayer MoS$_2$. The quantum well structures, using a stack of barrier/semiconductor/barrier as the building block, exhibit enhanced photoluminescence (PL) emission with each additional stack, indicating we are able to tune the light-matter coupling of 2D semiconducting TMDs[42,43]. We first confirm the formation of $N = 2$ aBN/MoS$_2$ superlattice after the first dry transfer step using cross-sectional HAADF-STEM and the corresponding EDS mapping of Mo, S, N, and Al signals (Fig. 3e). The estimated thicknesses from STEM images are 0.7 and 2.5 nm for the MoS$_2$ and aBN layers, respectively, validating the separation of monolayer MoS$_2$ with ultrathin aBN dielectric layers. The structural integrity of the monolayer MoS$_2$ after aBN encapsulation can also be confirmed by the observed $E_{2g}$ and $A_{1g}$ Raman modes of MoS$_2$ at 386.5 cm$^{-1}$ and 404.5 cm$^{-1}$, respectively, for the $N = 1$ stack. As $N$ increases, the Raman intensity increases and the FWHM narrows for $N > 1$ stacks due to strain release of the monolayer MoS$_2$ from the transfer process (Fig. 3f and Supplementary Fig. 7). The preservation of monolayer MoS$_2$ characteristics is also verified by the peak separation between $E_{2g}$ and $A_{1g}$, which remains constant at 19.7 cm$^{-1}$, a stark difference from multilayer MoS$_2$ where the peak separation increases linearly as layer number increases (Fig. 3f, inset). The increasing trend in peak separation is calculated from the Raman peaks of our MOCVD-grown multilayer MoS$_2$ (Supplementary Fig. 8) and is similarly observed in exfoliated MoS$_2$ flakes[44]. As a result of the monolayer MoS$_2$ confinement, increasing the repeating unit $N$ leads to a linear increase in the PL intensity (Fig. 3g). This points to an improvement in the probability of carrier recombination via dielectric integration with aBN, contrary to multilayer MoS$_2$ where PL intensity decreases with increasing layer number (Fig. 3g, inset) due to direct-to-indirect bandgap transition[45]. Once again, the $N = 1$ PL peak displays peak broadening with an excitonic wavelength of 681.0 nm, or 1.82 eV, due to substrate strain effects of as-grown monolayer MoS$_2$. Following the dry transfer and stacking, the PL spectra all remain at 1.86 eV from $N = 2-4$, consistent with other reports of monolayer MoS$_2$ at 1.85 eV, with no change in the electronic structure of MoS$_2$ from direct-to-indirect bandgap due to quantum well formation[46]. This demonstration shows that ALD aBN with controllable thickness can be applied to scalable TMD superlattice fabrication in place of CVD-grown hBN, Al$_2$O$_3$ dielectric, or WO$_x$[42,43].

## Transport properties of aBN-encapsulated MoS$_2$ FETs

Near ideal transport characteristics are observed in ALD aBN-encapsulated monolayer MoS$_2$ field effect transistors (FETs). We fabricate sets of double-gated ML MoS$_2$ FETs with aBN/HfO$_2$ stacks as gate dielectrics. Schematic illustrations of FETs at different process stages are shown in Fig. 4a): (i) On the left: back-gated ML MoS$_2$ FET with HfO$_2$ gate dielectric; On the right: back-gated ML MoS$_2$ FET with an aBN/HfO$_2$ stack as gate dielectric; (ii) an aBN layer deposited on top of the back-gated ML MoS$_2$ FET with an aBN/HfO$_2$ stack as gate dielectric; and (iii) a double-gated ML MoS$_2$ FET formed by depositing HfO$_2$ then top gate metal onto the stack shown in (ii). Because of the relatively small bandgap of aBN and potential for gate leakage, a stack of aBN/HfO$_2$ (3.6 nm/5.5 nm) is used to enable high carrier densities in the MoS$_2$ channel. The transfer curves of FETs with different channel lengths are shown in Fig. 4b. At $V_{ds} = 1.0$ V, the threshold voltage ($V_{th}$) shows little variation from device to device. The on-current level exhibits a clear dependence on channel length as shown in the linear plot corresponding to the right $y$-axis. Drain induced barrier lowering (DIBL) of only 55 mV/V at 1 nA is observed for a $L_{ch} \backsim 800$ nm ML MoS$_2$ FET, as shown in Fig. 4c, and the device shows a good on/off current ratio of up to 10$^9$. Figure 4d presents the full range subthreshold slope (SS) versus $I_{ds}$ for all devices characterized, again showing small device-to-device variations. The minimum SS for all channel lengths is close to 60 mV/dec at 10$^{-5}$ μA/μm, which indicates good electrostatic control from the back gate

and limited amount of trap charges from the bottom dielectric layer, not exceeding 10$^{12}$ cm$^{-2}$. By fabricating double-gated transistors, we also proved that aBN can be used as a top intermediate layer for subsequent HfO$_2$ deposition. Previous studies using a metal oxide seeding layer on MoS$_2$ showed significant off-state degradation due to trapped charges at the MoS$_2$ interfaces induced by oxygen defects in the metal oxide seeding layer[47,48]. Benefiting from the ALD growth of aBN, defect states are well controlled. Figure 4e shows the performance of an exemplary dual-gate controlled ML MoS$_2$ FET for the same device before and after top gate formation. The double gate structure enables an about two times higher current level compared with the "only back gate" structure as expected for the increased oxide capacitance. Using the same sweep range of $V_{BG}$, $V_{th\_cc}$ can be continuously changed by the top gate. The extracted $V_{BG\text{-th}}$ versus $V_{TG}$ dependence at 1 nA/μm is plotted in the inset of Fig. 4f showing a slope of about -0.5. Theoretically, the slope should be equal to the capacitance ratio between the top and the bottom dielectric. Since both the top dielectric and the bottom dielectric have the same dielectric stack with 3.6 nm aBN and 5.5 nm HfO$_2$, the expected slope of $V_{BG\text{-th}}/V_{TG}$ is −1. The smaller experimental slope implies a smaller total capacitance of the top gate dielectric stack, which is likely related to interface traps at the top interface/s, since the subthreshold slope as a function of top gate voltage alone is found to be significantly larger than the one as a function of back gate voltage alone, as shown in Supplementary Fig. 9. More research is required to fully understand this phenomenon. To further characterize the impact of aBN as an interfacial layer, it is important to carefully study its impact on the $V_{th}$ and SS, since previous reports have shown deteriorated SS and negatively shifted $V_{th}$ for both AlO$_x$ and TaO$_x$ seeding layers[49]. From Fig. 4g, we observe a slightly positive shift of $V_{th\_cc}$ after aBN deposition, and after the top gate formation, the $V_{th\_cc}$ exhibits a larger threshold voltage variation compared with only back gated devices, which is likely related to the higher interface trap density at the top gate as discussed above. Compared with the HfO$_2$ only substrates, the HfO$_2$/aBN dielectric stack has a much narrower distribution of threshold voltages, again attesting to the benefits of employing aBN as an interfacial layer between HfO$_2$ and the TMD. For the SS, as shown in Fig. 4h, there is no deterioration after aBN deposition and after top gate formation. (Note that in case of (iii), both gates are tied together, which compensates for the above discussed deteriorated SS when the top gate alone is used). In fact, the HfO$_2$/aBN dielectric stack allows for better SS, if compared with the HfO$_2$ only substrates. In Fig. 4i, we also compare the hysteresis distribution of the HfO$_2$ only and HfO$_2$/aBN stack, including after aBN deposition and after top gate formation. We find that devices fabricated on the HfO$_2$/aBN stack show a very small hysteresis and narrow distribution among devices. However, after the deposition of the top aBN interfacial layer, the devices exhibit larger hysteresis due to deposition-induced charges. This is not inconsistent with the findings in Fig. 4h, since in this figure, the SS values for (ii) were extracted only from the "positive-to-negative" gate voltage sweep. The much-improved gate control in case (iii) allows to recover the small hysteretic behavior as expected.

In conclusion, we have developed a wafer-scale, low-temperature ALD process for atomically smooth and conformal aBN dielectric for integration with 2D material-based electronics. We established the deposition temperature as a critical factor in controlling the aBN chemical composition, which in turn dictates the dielectric response of aBN. Quantum confinement of MoS$_2$ with aBN leads to enhanced Raman and PL intensities that increase linearly with the stacking number in scalable MoS$_2$/aBN quantum well structures. Additionally, we have demonstrated that ALD aBN serves as an interfacial layer for monolayer MoS$_2$ transistors, delivering excellent off-state performance, including a close-to-ideal subthreshold slope and minimal threshold voltage variations. We successfully demonstrated a double-

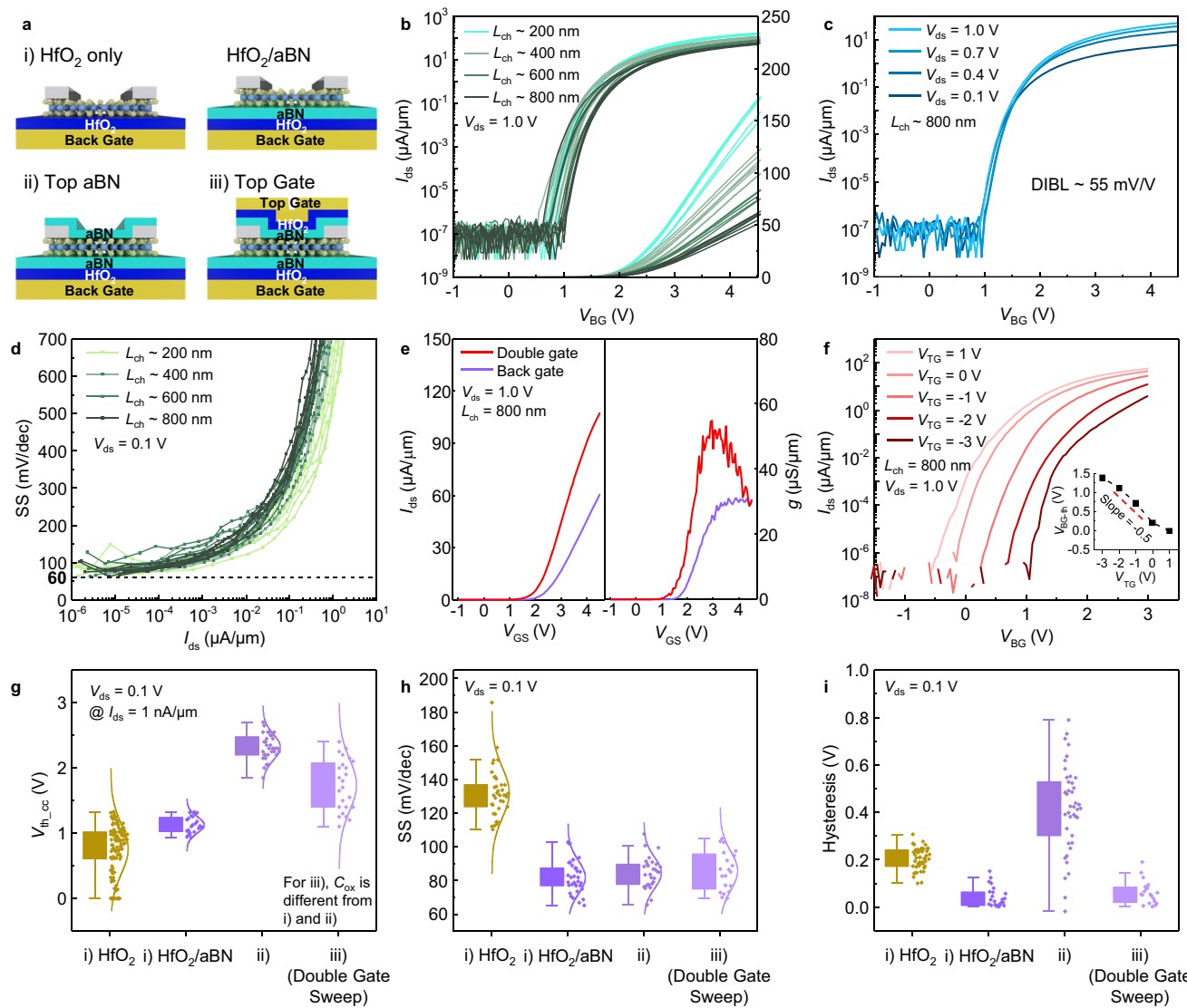

**Fig. 4 | Amorphous boron nitride encapsulated monolayer MoS$_2$ FETs.**
**a** Schematic representation of (i) back-gated ML MoS$_2$ FET with HfO$_2$ (left) and an aBN/HfO$_2$ stack (right) as gate dielectrics, (ii) the aBN layer on top of MoS$_2$ FETs shown on the right in i), and iii) the double-gated ML (monolayer) MoS$_2$ FET.
**b** Transfer characteristics of ML MoS$_2$ FETs with different channel lengths on aBN/HfO$_2$ substrate (back gate only as illustrated in (**a**) (i) on the right). **c** Transfer curves of a ML MoS$_2$ FET with $L_{ch}$ ~ 800 nm at different $V_{ds}$ (see (**a**) (i) on the right). **d** Full range SS versus $I_{ds}$ for different channel length FETs (see (**a**) (i) on the right).

**e** Transfer curves and transconductance plots of a ML MoS$_2$ FET (back gate versus double gate). **f** Transfer characteristics of a top-gate modulated ML MoS$_2$ FET. The inset shows the threshold voltage $V_{BG\_th}$ versus $V_{TG}$. **g** Distribution of $V_{th\_cc}$ at constant current for four types of ML MoS$_2$ FETs shown in (**a**); From left to right: (i) ML MoS$_2$/HfO$_2$/back gate, (i) ML MoS$_2$/aBN/HfO$_2$/back gate, (ii) aBN/ML MoS$_2$/aBN/HfO$_2$/back gate, and (iii) top gate/HfO$_2$/aBN/ML MoS$_2$/aBN/HfO$_2$/back gate. **h** SS-distribution of ML MoS$_2$ FETs. **i** Hysteresis distribution of ML MoS$_2$ FETs.

gated monolayer MoS$_2$ transistor encapsulated by ALD aBN, exhibiting excellent performance specs. Overall, our work paves a route towards scalable integration of dielectrics for the realization of advanced 2D-based electronics.

## Methods

### Substrate preparation
Si substrates (University Wafer, Inc.) are cleaned by sonication in acetone, isopropyl alcohol, and DI water for 10 min each. SiO$_2$ (10 μm)/Si substrates (University Wafer, Inc.) for trench patterning are cleaned with PRS-3000 for 10 min at 60 °C, followed by room temperature rinse in isopropyl alcohol and DI water for 2 min each. After pre-baking at 180 °C for 5 min, MaN-2405 resist is spin coated on the sample at 3000 rpm for 45 s and soft-baked at 90 °C for 2 min. Pattern is exposed using electron beam lithography (Raith EBPG-5200) with a dose of 700 μC/cm$^2$, then developed in CD-26 for 2 min and rinsed with DI water. For enhanced etch selectivity of SiO$_2$, 150 nm of Cr hard mask is deposited via evaporation (Temescal F-2000), followed by lift-off using PRS3000. SiO$_2$ is etched with CF$_4$ (Plasma-Therm Versalock700), resulting in a trench depth of 500 nm. Cr is removed with Cr etchant 1020 at room temperature for 15 min, followed by DI water rinse. Prior to aBN deposition, patterned SiO$_2$/Si substrates are cleaned by sonication in acetone, isopropyl alcohol, and DI water for 10 min each.

### ALD aBN synthesis
The ALD of BN is performed in a Kurt J. Lesker ALD150LX perpendicular-flow reactor. An Ebara ESR20N dry pump (nominal pumping speed: 46–70 cfm, base pressure: 15 mTorr) is used to pump down the chamber. All upstream ports are sealed by either metal seals or differentially pumped O-ring seals to minimize air permeation as described in ref. 50. 99.999% pure Ar is used as a carrier gas, which is

further purified by passing through an inert gas purifier (Entegris Gatekeeper GPU70 316LSS). The process pressure is maintained at ∽1.0 Torr. $BCl_3$ (Praxair, 99.999%) and $NH_3$ (Linde, 99.995%) are used for B and N sources, respectively. $NH_3$ is purified by passing through a MicroTorr MC1-703FV purifier. Two stage step-down delivery systems are built for both $BCl_3$ and $NH_3$ to reduce the dosing pressure to about 50 mTorr above the process pressure. Static dosing is performed for both $BCl_3$ and $NH_3$ precursors, including pulse and soak steps with no active pumping. The ALD cycle consists of the following six steps: (1) 1 s $BCl_3$ pulse, (2) 2 s soak, (3) 10 s purge, (4) 2 s $NH_3$ pulse, (5) 2 s soak, (6) 60 s purge. The Ar carrier gas flows for pulse and purge steps are 15 and 25 sccm, respectively.

## MOCVD $MoS_2$ growth
Monolayer $MoS_2$ material for $aBN/MoS_2$ quantum well structures is grown via MOCVD. A custom-built MOCVD system is utilized to grow monolayer $MoS_2$ films on 1 $cm^2$ c-plane sapphire substrates (Cryscore Optoelectronic Ltd, 99.996%). $Mo(CO)_6$ (99.99%, Sigma-Aldrich) serves as the Mo precursor and $H_2S$ (99.5%, Sigma-Aldrich) provides sulfur during synthesis. Details of the growth process for achieving uniform monolayer $MoS_2$ films have been previously described in our publication[51].

## Layer transfer and quantum well fabrication
aBN deposition on $MoS_2$ consists of 40 cycles of 65 °C aBN ALD followed by 225 cycles of 250 °C aBN ALD. To transfer the $aBN/MoS_2$ films and fabricate quantum well structures, the samples are first coated with a layer of PMMA (Micro Chem. 950 K A6) using a two-step spin casting process (step 1: 500 rpm for 15 s; step 2: 4500 rpm for 45 s). The PMMA/aBN/$MoS_2$ stack is then baked at 90 °C for 10 min. Next, thermal release tape (TRT) (Semiconductor Corp., release temperature 120 °C) is carefully placed on the top PMMA layer. The stack is then transferred to a DI water sonication bath for 15 min, facilitating the separation of the growth substrate. Subsequently, the TRT/PMMA/aBN/$MoS_2$ stack is thoroughly dried with $N_2$ to eliminate any water present at the interface prior to careful transfer onto the target substrate. After ensuring proper adhesion to the target substrate, the TRT is removed by heating it above its release temperature on a hot plate at 130 °C. The PMMA layer is removed by dissolving in acetone for 8 h. The sample is then rinsed in IPA and dried on a hot plate at 80 °C. The entire transfer process is repeated $N$ times to realize $(N + 1)$ aBN/$MoS_2$ quantum well structures.

## Spectroscopic ellipsometry
Film thickness and refractive index are evaluated using spectroscopic ellipsometry (J. A. Woollam M-2000) carried out from 192 to 1000 nm (Supplementary Fig. 3d–h). The Cauchy model is used to extract the thickness and refractive index of the BN layer.

## X-ray photoelectron spectroscopy
Chemical composition is measured by x-ray photoelectron spectroscopy (Physical Electronics VersaProbe II) with Al $K_\alpha$ monochromatic excitation source and a spherical sector analyzer. All spectra are charge corrected to C 1$s$ spectrum at 284.8 eV.

## Raman spectroscopy and photoluminescence
Raman and PL spectra are obtained using the Horiba Scientific LabRam HR Evolution VIS-NIR instrument with a laser wavelength of 532 nm at 3.4 mW for 10 s and 0.34 mW for 5 s, respectively.

## Transmission electron microscopy
The microstructures of the cross-sectional samples are observed by FEI Titan3 G2 double aberration-corrected microscope at 300 kV. All scanning transmission electron microscope (STEM) images are collected by using a high-angle annular dark field (HAADF) detector which has a collection angle of 52–253 mrad. EDS elemental maps of the sample are collected by using a SuperX EDS system under STEM mode. The electron energy loss spectroscopy (EELS) is performed under STEM mode by using a GIF Quantum 963 system. Thin cross-sectional TEM specimens are prepared by using focused ion beam (FIB, FEI Helios 660) lift-out technique. A thick protective amorphous carbon layer is deposited over the region of interest then Ga+ ions (30 kV then stepped down to 1 kV to avoid ion beam damage to the sample surface) are used in the FIB to make the samples electron transparent. The plane-view TEM (high resolution TEM (HRTEM) imaging, and the corresponding selected area electron diffraction (SAED) patterns) is performed on a FEI Talos F200x TEM at 200 kV.

## Absorbance measurement
The absorbance spectrum is measured using an Agilent/Cary 7000 spectrophotometer for aBN directly deposited on double-side polished transparent quartz substrates (University Wafer, Inc.).

## X-ray reflectivity
X-ray reflectivity (XRR) spectra of aBN deposited on Si are collected using Malvern Panalytical X'Pert3 MRD with Cu $K_\alpha$ source at 40 mA and 45 kV. Spectra are fitted using the X'Pert Reflectivity software.

## Atomic force microscopy (AFM)
AFM is conducted using Bruker Dimension Icon instrument with a RTESPA-150 tip in peak-force tapping mode. AFM scans are collected with a peak force setpoint of 1.00 nN and a scan rate of 1.00 Hz.

## Dielectric constant and breakdown voltage measurements
Dielectric characterization of aBN is carried out on Pt/aBN/p-Si device structures. p-Si substrate is treated with buffered oxide etch (BOE 6:1) for 5 min to remove surface native oxide, then rinsed with DI water and dried with $N_2$ gas. The p-Si substrate is immediately transferred to the ALD chamber for direct deposition of aBN. A shadow mask with 200 μm diameter holes is used to sputter 200 nm thick Pt electrodes onto aBN/p-Si. $C–f$ measurements of Pt/aBN/p-Si are measured from 1 kHz to 1 MHz at 30 mV using Agilent HP4980 Precision LCR Meter. $I–V$ characteristics are measured using the HP4140B pA meter.

## Double-gated monolayer $MoS_2$ FET fabrication and analysis
Monolayer $MoS_2$ material for FET fabrication is purchased from 2D semiconductors. ML $MoS_2$ single crystals are wet transferred on to the local bottom gate substrates. The local bottom gate substrates consist of a stack of Cr/Au (2/13 nm) as gate metal and 5.5 nm $HfO_2$ and 3.6 nm aBN as dielectric layer grown by ALD. aBN deposition consists of 150 cycles of 250 °C aBN ALD. After the wet transfer process, the sample is annealed at a pressure of ∽5 × $10^{-8}$ Torr at 200 °C for 2 h. Optical microscopy is used to identify flakes located on local bottom gates and the sample is spin coated with photoresist PMMA and baked at 180 °C for 5 min. Next, source/drain contacts are patterned by a JEOL JBX-8100FS E-Beam Writer system and developed in an IPA/DI solution followed by e-beam evaporation of 70 nm Ni as contact metal. Next, the sample undergoes a lift-off process. To fabricate a top gate hybrid stack of aBN/$HfO_2$, 3.6 nm aBN and 5.5 nm $HfO_2$ are deposited by ALD. The top aBN deposition consists of 40 cycles of 65 °C aBN ALD followed by 90 cycles of 200 °C aBN ALD. Finally, the top gate metal is defined by e-beam lithography (EBL), and e-beam evaporation of 50 nm Ni, followed by another lift-off process in acetone. Details of the FET fabrication process flow have been described previously[47]. A Lake Shore CPX-VF probe station and Agilent 4155C Semiconductor Parameter Analyzer are used to perform the electrical characterization at room temperature in high vacuum (≈$10^{-6}$ Torr). Standard DC sweeps are used in the electrical measurements for all devices. All devices are measured as fabricated.

## Data availability

The authors declare that the data supporting the findings of this study are available within the paper and Supplementary Information. Extra data are available from the corresponding authors upon request.

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

## Acknowledgements

C.Y.C., R.T., Y.-C.L., and J.A.R. acknowledge funding from NEWLIMITS, a center in nCORE as part of the Semiconductor Research Corporation (SRC) program sponsored by NIST through award number 70NANB17H041. Y.-C.L. acknowledges the support from the Center for Emergent Functional Matter Science (CEFMS) of National Yang Ming Chiao Tung University and the Yushan Young Scholar Program from the Ministry of Education of Taiwan. C.Y.C. and J.A.R. acknowledge Intel through the Semiconductor Research Corporation (SRC) Task 2746, the Penn State 2D Crystal Consortium (2DCC)-Materials Innovation Platform (2DCC-MIP) under NSF cooperative agreement DMR-1539916, and NSF CAREER Award 1453924 for financial support. CVD hBN reference for this publication was provided by The Pennsylvania State University Two-Dimensional Crystal Consortium – Materials Innovation Platform (2DCC-MIP), which is supported by NSF cooperative agreement DMR-1539916.

## Author contributions

C.Y.C. conducted the ALD film growth, characterization, and data analysis. Z.S. performed FET device fabrication and electrical measurements. J.A. and Z.C. supervised the FET analysis and discussions. R.T. conducted the CVD of 2D $MoS_2$ and film transfer experiments. Y.-C.L. conducted electrical characterization with C.Y.C., developed quantum well stack with R.T., and participated in data analysis. K.W. performed the TEM experiments. B.L. and G.B.R. helped with ALD tool and process development. J.K., Y.-C.L., and J.A.R. supervised the project. All authors participated in the manuscript review.

## Competing interests

The authors declare no competing interests.
