## [Peer Review File · Nature Communications]

Tailoring Amorphous Boron Nitride for High-Performance 2D ElectronicsREVIEWER COMMENTS

Reviewer #1 (Remarks to the Author):

The current manuscript delineates the ALD-based a-BN growth techniques and its applicability in the field of 2D electronics. The structural/chemical properties of aBN materials in relation to growth temperature have been systematically analyzed, and strategies for their implementation in 2D electronics have been clearly elucidated. In light of the significant efforts being expended in the pursuit of gate dielectric and interface materials for 2D electronics, it is envisaged that the outcomes of this study will draw the attention of the relevant community. Nevertheless, for this manuscript to be deemed suitable for publication in Nature Communications, I recommend providing further clarification and supplementation on the following queries.

To determine the cause of the influx of a significant amount of oxygen at low temperatures, the authors produced and analyzed a hybrid a-BN film that was grown sequentially at low and high temperatures. However, there are still unclear elements in claiming that the a-BN thin film grown at 65 degrees does not deform during the subsequent a-BN thin film growth process at 200 degrees.

(Q1) Even if an aBN sample grown at 65 degrees undergoes subsequent post-treatment at 200 degrees in the chamber, will it have oxygen contents like the as-grown sample without heat treatment?

(Q2) If the aBN sample grown at a low temperature of 65 degrees reacts with atmospheric oxygen, it is thought that the reaction will be concentrated on the surface. If you grow a thick a-BN sample and analyze the atomic concentration in the depth direction, will there be a change? Or is there a difference depending on the thickness of the aBN thin film in the oxygen level measured by the surface?

(Q3) Why is the oxygen content of the hybrid a-BN film lower at 3% than the 4% of the thin film grown at 200 degrees?

A trench pattern with 10 x 10 μm^2 area and 500 nm depth was used to demonstrate conformal growth behavior. However, in order to more clearly assert the advantages of ALD, the top/side/bottom situation must be shown in a trench pattern with a higher aspect ratio.

(Q4) Have you tested in a high aspect ratio structure?

The authors claim that hBN can form a clean vdW interface with the 2D semiconducting channel. It can substantially reduce charge carrier scattering due to surface roughness and charged impurities, also leads to reduced remote phonon scattering since the high energy surface optical phonon modes of hBN do not couple to the low energy modes in 2D semiconductors. However, a-BN has structural/physical properties that are significantly different from h-BN. It has no vdW interface and even dielectric constant is not high as much as the others, such as HfO₂ and Al₂O₃.

(Q5) Nevertheless, is there any basis for judging that it is useful as a dielectric in 2D electronic devices?

(Q6) Two step approach at different temperature is applied to deposit aBN on MoS₂. What happens if 250 degree aBN is deposited directly on the MoS₂ surface without low-temperature aBN deposition?

(Q7) What is the minimum thickness of aBN before HfO₂ deposition for MoS₂ device fabrication?

Reviewer #2 (Remarks to the Author):

Chen et al. have reported on the growth of Amorphous Boron Nitride (aBN), highlighting its potential applications in post-Moore era electronics. It is commendable to approach material growth with a specific focus on electronic devices and circuits. However, despite this perspective being reiterated in the title, abstract, and introduction, the paper lacks a clear demonstration of how this material growth directly contributes to the enhancement of device parameters. For high-performance digital electronic devices, particularly ultra-scaled short-channel devices in advanced technical nodes below 10 nm, it is crucial. Yet, the devices examined in this study still operate within the diffusive regime. Furthermore, 'high-performance' should also imply superior transistor characteristics, such as high on-state current and fast switching with low power consumption (intrinsic gate delay and energy-delay product). Regrettably, these aspects are overlooked in the paper, weakening the claim of 'high-performance.'

Additionally, I have several concerns which I outline below:

1. What is the defect density in aBN? Can the authors provide spectroscopy data to quantify the defect condition and explain how these defects impact transistor behavior? Also, can the interfacial trap state density and border trap density be quantified?
2. The gate control efficiency is influenced by the thickness of the aBN dielectric. Can the authors compare the efficiency of single-layer or bilayer aBN with high-k dielectric in terms of gate efficiency improvement?
3. Regarding the extraction of the SS values, its significance diminishes as the current approaches the off-state. As shown in Figure 4d, at an off-state current of 100 nA/um, the SS value is 200 mV/dec, far from the "Near ideal transport characteristics" claimed for ALD aBN-encapsulated monolayer MoS2 FETs.
4. Could the authors elaborate on achieving "trap charges from the bottom dielectric layer, not exceeding 10^{12} cm^{-2} "?
5. Providing additional experimental evidence for the abnormal k values of the aBN grown at 65 degrees Celsius would be beneficial.

Reviewer #3 (Remarks to the Author):

Cindy Y. Chen et al. demonstrated a wafer-scale, low-temperature ALD process of aBN for integration with 2D material-based devices. The authors controlled the chemical composition of aBN by changing the deposition temperature from 65 to 250 oC. They fabricated not only the quantum well structure (barrier/semiconductor/barrier) but double-gated FET structures using aBN interfacial layered HfO2 gate dielectric. As a result, their device performances considerably improved.

Interestingly, the intrinsic dielectric properties of ALD aBN do not show low dielectric constant. They varied from 4.3 to 8.6 at 100 kHz, depending on the deposition temperature. This result conflicts with previously reported aBN literature [REF 37-40 in original manuscript]. Therefore, it is necessary to define the differences compared to reported aBN structures. In this aspect, I believe that this work may be considered for publication after correcting somewhat misleading statements.

1. Why does aBN show a relatively high dielectric constant (k)? They have higher k than crystalline BN (~ 3.2). For example, if their deposition temperature exceeds 300 oC, does k value decrease?
2. In Figure 1b, they measured the Raman spectrum of aBN on a Si substrate to confirm a non-crystalline BN structure. This reviewer recommends adding the aBN Raman spectrum measured on the SiO2 substrate because detecting the E2g signal of thin crystalline BN is difficult on the Si substrate. Furthermore, many characterizations of aBN are conducted on SiO2 in this study. The oxide layer can affect the crystallinity of BN during deposition.
3. To clarify the thickness of the aBN film, the authors have to show the height profiles of their aBN films as a function of deposition temperature using atomic force microscopy.
4. In the aBN/MoS2 quantum well stack structure, what is the difference between hybrid ALD (65

oC and 250 oC) and 250 oC ALD for a device?

5. In Figure 3e, the EDS map of N is unclear to readers. It should be revised to a clear image.

RESPONSE TO REVIEWERS' COMMENTS

Reviewer #1 (Remarks to the Author):

The current manuscript delineates the ALD-based a-BN growth techniques and its applicability in the field of 2D electronics. The structural/chemical properties of aBN materials in relation to growth temperature have been systematically analyzed, and strategies for their implementation in 2D electronics have been clearly elucidated. In light of the significant efforts being expended in the pursuit of gate dielectric and interface materials for 2D electronics, it is envisaged that the outcomes of this study will draw the attention of the relevant community. Nevertheless, for this manuscript to be deemed suitable for publication in Nature Communications, I recommend providing further clarification and supplementation on the following queries. To determine the cause of the influx of a significant amount of oxygen at low temperatures, the authors produced and analyzed a hybrid a-BN film that was grown sequentially at low and high temperatures. However, there are still unclear elements in claiming that the a-BN thin film grown at 65 degrees does not deform during the subsequent a-BN thin film growth process at 200 degrees.

(Q1) Even if an aBN sample grown at 65 degrees undergoes subsequent post-treatment at 200 degrees in the chamber, will it have oxygen contents like the as-grown sample without heat treatment?

Our experimental results from the hybrid ALD aBN film stack suggest that the 200 °C aBN ALD transforms the 65 °C aBN component film via two proposed mechanisms: 1) increased extent of reaction between B-Cl and N-H components in the 65 °C aBN film through the elevated temperature and 2) increased extent of reaction with incoming precursors. Both mechanisms likely play a role in the transformation of 65 °C film during the hybrid ALD process, and suggest that a post-treatment of 200 °C thermal annealing will further enhance the unreacted B-Cl and N-H towards stoichiometric aBN. Thus, although post-annealing has not been tested, we expect the post-treatment at 200 °C to result in lower overall O content.

(Q2) If the aBN sample grown at a low temperature of 65 degrees reacts with atmospheric oxygen, it is thought that the reaction will be concentrated on the surface. If you grow a thick a-BN sample and analyze the atomic concentration in the depth direction, will there be a change? Or is there a difference depending on the thickness of the aBN thin film in the oxygen level measured by the surface?

As the data suggests oxidation of unreacted B-Cl bonds upon air exposure, oxidation is thought to occur initially from the surface to the bulk film. Based on this finding, we expect immediately upon air exposure, a thicker 65 °C aBN film should initially have B-O components concentrated on the surface, and extend towards the bulk as the time exposed to ambient increases.

(Q3) Why is the oxygen content of the hybrid a-BN film lower at 3% than the 4% of the thin film grown at 200 degrees?

This is within the typical observed deviation range of XPS analysis of the aBN films. The XPS spectra of the two aBN films (pure and hybrid) and their fitted components do not indicate a significant shift or change in intensity (**Figure R1.1**).

Figure R1.1: B 1s XPS spectra and component fits for pure 200 °C and hybrid (65 °C followed by 200 °C) aBN films.

A trench pattern with 10x10 μm^2 area and 500 nm depth was used to demonstrate conformal growth behavior. However, in order to more clearly assert the advantages of ALD, the top/side/bottom situation must be shown in a trench pattern with a higher aspect ratio.

(Q4) Have you tested in a high aspect ratio structure?

We thank the reviewer for the suggestion. We have added the top/side/bottom image in the SI (**Supplementary Figure 2**). As we have not tested ALD on a higher aspect ratio structure, we have modified the text (**Main Text**, page 3 and 5) to emphasize conformality and uniform step coverage on structured surfaces rather than high aspect ratio structures.

The authors claim that hBN can form a clean vdW interface with the 2D semiconducting channel. It can substantially reduce charge carrier scattering due to surface roughness and charged impurities, also leads

Joshua A. Robinson, Professor

Department of Materials Science and Engineering
Department of Physics
Department of Chemistry
NSF-I/UCRC Atomically Thin Multifunctional Coatings
NSF-MIP 2D Crystal Consortium

The Pennsylvania State University
N-337 Millennium Science Complex
University Park, PA 16802
jrobinson@psu.edu; (814) 863-8567

to reduced remote phonon scattering since the high energy surface optical phonon modes of hBN do not couple to the low energy modes in 2D semiconductors. However, a-BN has structural/physical properties that are significantly different from h-BN. It has no vdW interface and even dielectric constant is not high as much as the others, such as HfO_2 and Al_2O_3 .

(Q5) Nevertheless, is there any basis for judging that it is useful as a dielectric in 2D electronic devices?

From our experimental results, aBN is a better seeding layer compared with other oxide-based seeding layer, such as AlO_x and TaO_x ^{1,2}. Despite a strong (desired) n-doping effect that is a result of the use of oxide seeding layers, a substantial degradation of the inverse subthreshold slope and off-state current is detrimental for any device performance. In our work, when using aBN as an interfacial layer to form the top gate, both high on-state currents and good off-state performance are achieved simultaneously.

Previous works indicate that FET of graphene on aBN shows performance improvement in FET mobility and carrier homogeneity and reduced extrinsic doping in graphene compared to graphene FET with SiO_2 -only dielectric. This improvement can be attributed to reduced ambient- and substrate-induced doping and dielectric screening of charged impurities.³⁻⁴ In addition, aBN was proved suitable as a passivation and heat passivation layer on top of TMDs⁵ or between TMDs and oxide dielectric (where aBN can reduce interfacial thermal resistance)⁶ that helps improve TMD's FET performance. While its relatively low k doesn't help push for a smaller device EOT, Ref.³⁻⁶ and our ALD-grown aBN show that aBN can be useful in 2D electronic devices with different approaches. Finally, recent theoretical and experimental studies indicate that aBN alone is a highly effective diffusion barrier for Cu and suitable for nanoelectronic integration.⁷⁻⁸ As more people are looking into the back-end integration and low-temperature synthesis for 2D semiconductors, aBN on TMDs like what we demonstrate in this manuscript will play an important role down the roads.

Another technology-related point in favor of aBN is that future 2D-based nano-sheet FETs are expected to be used in conjunction with a gate-all-around (GAA) geometry. In this context, an ALD-grown amorphous BN will be beneficial as seeding layer for ML MoS_2 GAA FETs, since it can be uniformly deposited everywhere on the MoS_2 layer. Directional e-beam deposition of an oxide-based seeding layer is not an option in this case due to the inability to cover the "back side" of the nano-sheet channel.

(Q6) Two step approach at different temperature is applied to deposit aBN on MoS_2 . What happens if 250 degree aBN is deposited directly on the MoS_2 surface without low-temperature aBN deposition?

We show that a one-step aBN deposition at 250 °C on MoS_2 yields a discontinuous aBN film with nucleation occurring preferentially along the MoS_2 grain boundaries (**Supplementary Figure 5a**). This is further quantified and supported by XPS analysis, in which aBN film deposited at 250 °C without the low-temperature step exhibits a B/S ratio ~ 0 (**Supplementary Figure 6c**).

(Q7) What is the minimum thickness of aBN before HfO₂ deposition for MoS₂ device fabrication?

40 ALD cycles is the minimum cycle number of low-temperature (65 °C) interfacial layer required for forming a continuous higher-temperature (200 – 250 °C) aBN film on MoS₂ surface. The corresponding minimum thickness of this interfacial layer is about 2.5 nm, such as the ones we used for the quantum well structures made of aBN/MoS₂ stacks. **Figure R1.2** is a close-view cross-sectional HAADF image that shows ~ 2.5 nm aBN thickness of an example from this recipe. The aBN before HfO₂ deposition for the top-gated MoS₂ device fabrication was slightly thicker, 3.6 nm, because we wanted to minimize dielectric pinhole formation and or pinhole density on the top MoS₂ device surface.

Figure R1.2: A HAADF image of an Au-encapsulated aBN film deposited on silica by our ALD.

References

1. McClellan, Connor J., Eilam Yalon, Kirby K. H. Smithe, Saurabh V. Suryavanshi, and Eric Pop. "High Current Density in Monolayer MoS₂ Doped by AlO_x." *ACS Nano* 15, no. 1 (2021): 1587–96. <https://doi.org/10.1021/acsnano.0c09078>.
2. H. -Y. Lan, V. P. Oleshko, A. V. Davydov, J. Appenzeller and Z. Chen, "Dielectric Interface Engineering for High-Performance Monolayer MoS₂ Transistors via TaO_x Interfacial Layer," in *IEEE Transactions on Electron Devices*, vol. 70, no. 4, pp. 2067-2074, April 2023, doi: 10.1109/TED.2023.3251965.
3. Uddin, Md Ahsan, et al. "Mobility enhancement in graphene transistors on low temperature pulsed laser deposited boron nitride." *Applied Physics Letters* 107.20 (2015).
4. Sattari-Esfahlan, Seyed Mehdi, et al. "Low-Temperature Direct Growth of Amorphous Boron Nitride Films for High-Performance Nanoelectronic Device Applications." *ACS Applied Materials & Interfaces* 15.5 (2023): 7274-7281.
5. Lu, Zhanjie, et al. "Low-temperature synthesis of boron nitride as a large-scale passivation and protection layer for two-dimensional materials and high-performance devices." *ACS Applied Materials & Interfaces* 14.22 (2022): 25984-25992
6. Liu, Donghua, et al. "Conformal hexagonal-boron nitride dielectric interface for tungsten diselenide devices with improved mobility and thermal dissipation." *Nature Communications* 10.1 (2019): 1188.
7. Kaya, Onurcan, et al. "Amorphous Boron Nitride as a Diffusion Barrier to Cu Atoms." *arXiv preprint arXiv:2402.01251* (2024).
8. Kim et al., "Ultralow-k Amorphous Boron Nitride Film for Copper Interconnect Capping Layer," in *IEEE Transactions on Electron Devices*, vol. 70, no. 5, pp. 2588-2593, May 2023, doi: 10.1109/TED.2023.3258403

Reviewer #2 (Remarks to the Author):

Chen et al. have reported on the growth of Amorphous Boron Nitride (aBN), highlighting its potential applications in post-Moore era electronics. It is commendable to approach material growth with a specific focus on electronic devices and circuits. However, despite this perspective being reiterated in the title, abstract, and introduction, the paper lacks a clear demonstration of how this material growth directly contributes to the enhancement of device parameters. For high-performance digital electronic devices, particularly ultra-scaled short-channel devices in advanced technical nodes below 10 nm, it is crucial. Yet, the devices examined in this study still operate within the diffusive regime. Furthermore, ‘high-performance’ should also imply superior transistor characteristics, such as high on-state current and fast switching with low power consumption (intrinsic gate delay and energy-delay product). Regrettably, these aspects are overlooked in the paper, weakening the claim of ‘high-performance’.

We do agree with the reviewer in general that ultra-scaled short-channel devices with channel lengths below 50 nm are important to ultimately achieve high on-current levels. However, as shown in **Figure R2.1** below, the contact resistance in our case is still in the thousands of ohm- μm range, far from the desired quantum limit of contact resistance. This implies that even if the device channel is scaled towards 10 nm, one does not expect any improvement in device performance, which is the reason that such a study has not been performed in this manuscript which focuses on the gate stack and not contacts.

Figure R2.1: Contact resistance of Ni contacted ML MoS₂ FETs

Joshua A. Robinson, Professor

Department of Materials Science and Engineering
Department of Physics
Department of Chemistry
NSF-I/UCRC Atomically Thin Multifunctional Coatings
NSF-MIP 2D Crystal Consortium

The Pennsylvania State University
N-337 Millennium Science Complex
University Park, PA 16802
jrobinson@psu.edu; (814) 863-8567

Instead, our focus has been on demonstrating superior off-state behavior, including good inverse subthreshold slopes and very small hysteresis behavior below a threshold. Our findings indicate that good electrostatic gate control has been achieved in our gate dielectric stack that involves aBN. As mentioned above, while higher on-current levels have indeed been achieved in double gate MoS₂ FETs, those devices showed a severely degraded, unacceptable off-state behavior.

We also note that ALD-grown aBN offers several process advantages over crystalline hBN for the continuous improvement of dielectrics in 2D electronics. Compared to the “transfer-required” CVD-grown hBN (or exfoliated hBN crystals), ALD-aBN is performed at significantly lower process temperatures (< 250 °C) and on a wafer-scale, thus providing a better approach towards future 2D electronic integration schemes. Another technology-related point in favor of aBN is that future 2D-based nano-sheet FETs are expected to be used in conjunction with a gate-all-around (GAA) geometry. In this context, an ALD-grown amorphous BN will be beneficial as seeding layer for ML MoS₂ GAA FETs, since it can be uniformly deposited everywhere on the MoS₂ layer. Directional e-beam deposition of an oxide-based seeding layer is not an option in this case due to the inability to cover the “back side” of the nanosheet channel.

Additionally, I have several concerns which I outline below:

1. What is the defect density in aBN? Can the authors provide spectroscopy data to quantify the defect condition and explain how these defects impact transistor behavior? Also, can the interfacial trap state density and border trap density be quantified?

One way of mapping out the defect density in aBN is the usage of STM spectroscopy. While we did not perform this type of study, our experimental transport data provide quantitative information about the interface trap density in our gate stack structures. From the inverse subthreshold slope SS in the device off-state, we extract an interfacial trap state density of $1.098 \times 10^{12} \text{ cm}^{-2}$. This value is extracted by noting that in the deep off-state that is dominated by thermal emission over the conduction band edge of MoS₂, $SS = 2.3 \cdot kT/q \cdot n$, where $n = 1 + (C_D + C_{it})/C_{ox}$. Noting that C_D is zero for a fully depleted device as ML MoS₂, $n = 1 + C_{it}/C_{ox}$ in our case. As shown in **Figure 4h in the main text** of the manuscript, SS is around or smaller than 80 mV/dec, which implies a C_{it} of $0.33 \cdot C_{ox}$.

To carefully extract $C_{ox} = C_{MOS}$ from the back-gated ML MoS₂ FETs, we measured a MOS capacitor (CAP) of ML MoS₂ on HfO₂/aBN dielectric (**Figure R2.2**), and determined an oxide capacitance of 0.523 uC/cm², which means that C_{it} is around 0.174 μF/cm², which equals to $1.086 \times 10^{12} \text{ eV}^{-1} \text{ cm}^{-2} \text{ V}^{-1}$

Figure R2.2. a) Representative SEM image of a ML MoS₂ MOSCAP. b) C-V characteristics of a ML MoS₂ MOSCAP with an area of 150 μm² and a gate dielectric stack of 3.6nm aBN+5nm HfO₂. c) Area-dependent capacitance plot.

The main goal of our ALD process is to produce pure amorphous BN – without any nanostructures at the atomic scale. Our TEM data in the paper provide clear evidence about the amorphous structure. Techniques such as STM/STS and Raman and CL spectroscopy commonly seen on qualifying hBN are not useful for our aBN. Our XPS analysis indicates that the B to N ratio is 0.93 ± 0.02 , which suggests our aBN films might be slightly boron deficient. Thus, this cation deficiency may contribute to the p-doping effect that prevents V_{TH} shift toward more negative due to charge interaction between amorphous oxide dielectric (HfO₂ in this study) and MoS₂ FET.

2. The gate control efficiency is influenced by the thickness of the aBN dielectric. Can the authors compare the efficiency of single-layer or bilayer aBN with high-k dielectric in terms of gate efficiency improvement?

As the aBN films are amorphous, they exhibit no crystalline order and are not layered structures. Here, we consider two aBN thickness values: 3.6 and 1.8 nm. The metal-insulator-metal capacitance – C_{MIM} – for our gate stack (3.6nm aBN+5nm HfO₂) has been experimentally determined to be 0.695 μF/cm². Due to the existence of an “air gap” between the dielectric and the ML MoS₂^{3,4}, the measured value for C_{MOS} is found to be always smaller than C_{MIM} . For the same 3.6 nm aBN+5 nm HfO₂ gate stack, we measured C_{MOS} to be around 0.57 μF/cm².

Once the dielectric film thickness for aBN is scaled down from 3.6 to 1.8 nm, the C_{MIM} of 1.8nm aBN + 5nm HfO₂ is expected to be 1.07 μF/cm². This means from **Figure R2.3** that C_{MOS} is expected to be around 0.8 μF/cm², 1.4 times larger than the value for the stack of 3.6 nm aBN + 5 nm HfO₂. (k of aBN is 3.9)

As a result, if we are scaling down the aBN thickness from 3.6 to 1.8 nm, we will create a 1.4 times larger carrier density in the MoS₂ channel for the same gate overdrive conditions.

Figure R2.3. a) C_{MOS} vs. C_{MIM} and simulated C_{MOS} -values assuming a 0.22nm air gap⁴

3. Regarding the extraction of the SS values, its significance diminishes as the current approaches the off-state. As shown in Figure 4d, at an off-state current of 100 nA/μm, the SS value is 200 mV/dec, far from the "Near ideal transport characteristics" claimed for ALD aBN-encapsulated monolayer MoS₂ FETs.

Please note that the SS value close to threshold is NOT determined by the thermal inverse subthreshold slope, but rather by the Schottky barrier heights (SBHs) at the source and drain electrode. Since we are using Ni as a contact material, the SBH is relatively large for our devices, which means that at an off-state current of 100nA/μm, the current is dominated by the SB branch of the SS, instead of the thermionic-limited branch. Please refer to reference [5] as an example for a Schottky barrier dominated FET from black phosphorus. To measure SS-values in the thermionic-limited branch, one needs to focus on current levels in the 10^{-5} ~ 10^{-4} μA/μm range.

4. Could the authors elaborate on achieving "trap charges from the bottom dielectric layer, not exceeding 10^{12} cm⁻²"?

The value of interface trap charges is calculated from the inverse subthreshold slopes. Please refer to the first question for the detailed calculation steps.

5. Providing additional experimental evidence for the abnormal k values of the aBN grown at 65 degrees Celsius would be beneficial.

At a deposition temperature of 65 °C, the aBN film is comprised of a high concentration of O based on XPS quantification using the B-O component (**Figure R2.4a**). In addition to B-O components, the 65 °C aBN also exhibits N-H components in the N 1s spectra (**Figure R2.4b**), which is indicative of unreacted N-H because of insufficient thermal energy for a complete reaction. We hypothesize that the anomalous dielectric constant is attributed to the observed higher incorporation of impurities at 65 °C.

Figure R2.4. High-resolution XPS spectra of aBN deposited at 65 °C in the a) B 1s and b) N 1s regions.

References

1. McClellan, Connor J., Eilam Yalon, Kirby K. H. Smithe, Saurabh V. Suryavanshi, and Eric Pop. "High Current Density in Monolayer MoS₂ Doped by AlOx." *ACS Nano* 15, no. 1 (2021): 1587–96. <https://doi.org/10.1021/acsnano.0c09078>.
2. H. -Y. Lan, V. P. Oleshko, A. V. Davydov, J. Appenzeller and Z. Chen, "Dielectric Interface Engineering for High-Performance Monolayer MoS₂ Transistors via TaOx Interfacial Layer," in *IEEE Transactions on Electron Devices*, vol. 70, no. 4, pp. 2067-2074, April 2023, doi: 10.1109/TED.2023.3251965.
3. Arutchelvan, G., Smets, Q., Verreck, D. et al. Impact of device scaling on the electrical properties of MoS₂ field-effect transistors. *Sci Rep* 11, 6610 (2021).
4. Z. Sun, C. Chen, J. A. Robinson, Z. Chen and J. Appenzeller, "A mobility study of monolayer MoS₂ on low- κ /high- κ dielectrics," 2023 Device Research Conference (DRC), Santa Barbara, CA, USA, 2023, pp. 1-2, doi: 10.1109/DRC58590.2023.10258241.
5. Haratipour, Nazila, Seon Namgung, Sang-Hyun Oh, and Steven J. Koester. "Fundamental Limits on the Subthreshold Slope in Schottky Source/Drain Black Phosphorus Field-Effect Transistors." *ACS Nano* 10, no. 3 (March 22, 2016): 3791–3800. <https://doi.org/10.1021/acsnano.6b00482>.

Joshua A. Robinson, Professor

Department of Materials Science and Engineering
Department of Physics
Department of Chemistry
NSF-I/UCRC Atomically Thin Multifunctional Coatings
NSF-MIP 2D Crystal Consortium

The Pennsylvania State University
N-337 Millennium Science Complex
University Park, PA 16802
jrobinson@psu.edu; (814) 863-8567

Reviewer #3 (Remarks to the Author):

Cindy Y. Chen et al. demonstrated a wafer-scale, low-temperature ALD process of aBN for integration with 2D material-based devices. The authors controlled the chemical composition of aBN by changing the deposition temperature from 65 to 250 °C. They fabricated not only the quantum well structure (barrier/semiconductor/barrier) but double-gated FET structures using aBN interfacial layered HfO₂ gate dielectric. As a result, their device performances considerably improved.

Interestingly, the intrinsic dielectric properties of ALD aBN do not show low dielectric constant. They varied from 4.3 to 8.6 at 100 kHz, depending on the deposition temperature. This result conflicts with previously reported aBN literature [REF 37-40 in original manuscript]. Therefore, it is necessary to define the differences compared to reported aBN structures. In this aspect, I believe that this work may be considered for publication after correcting somewhat misleading statements.

1. Why does aBN show a relatively high dielectric constant (κ)? They have higher κ than crystalline BN (~3.2). For example, if their deposition temperature exceeds 300 °C, does κ value decrease?

Amorphous and crystalline hexagonal boron nitride (hBN) both are reported to exhibit a range of dielectric constants depending on film characteristics and processing methods. In our study, we determined that deposition temperature plays a significant role in the O concentration of the aBN, which consequently impacts the determined aBN dielectric constant. Specifically, as O concentration increases with lower deposition temperature, the dielectric constant increases. The higher κ observed for 65 °C aBN in this study is markedly different from other reported κ of aBN, which are from near stoichiometric aBN, and therefore have minimized dipole contribution from polar B-O components. Based on these findings, we also expect that higher deposition temperatures, such as 300 °C, may further decrease the dielectric constant due to O concentration reduction in aBN. We also note that in this study, temperatures above 250 °C have not been explored in depth due to considerations of aBN as an interface dielectric for 2D applications and recrystallization that might occur at 300 °C. To minimize the 2D material structural damage, aBN is performed at or below the temperature of 250 °C.

2. In Figure 1b, they measured the Raman spectrum of aBN on a Si substrate to confirm a non-crystalline BN structure. This reviewer recommends adding the aBN Raman spectrum measured on the SiO₂ substrate because detecting the E_{2g} signal of thin crystalline BN is difficult on the Si substrate. Furthermore, many characterizations of aBN are conducted on SiO₂ in this study. The oxide layer can affect the crystallinity of BN during deposition.

We thank the reviewer for the suggestion. We have performed Raman spectroscopy of aBN on 90 nm SiO₂ but did not observe the E_{2g} of crystalline BN (**Figure R3.1a**). We have also observed a similar absence of the E_{2g} signal of thick (30 nm) aBN on Si (**Figure R3.1b**). Additionally, we note that the Si substrates used

in the aBN Raman characterization have surface native oxide, which presents similar -OH surface sites as in SiO₂ substrates for the ALD-BN reaction. Combined with the relatively low deposition temperature range in the ALD process (65 – 250 °C), aBN remains amorphous on both Si and SiO₂ substrates. It is worth noting that our ALD process can produce thick aBN (30 nm) without formation of hBN nanostructures that have been observed in 7 nm aBN grown by PECVD [See Figure 3 in “K. Kim *et al.*, "Ultralow-k Amorphous Boron Nitride Film for Copper Interconnect Capping Layer," in *IEEE Transactions on Electron Devices*, vol. 70, no. 5, pp. 2588-2593, May 2023, doi: 10.1109/TED.2023.3258403"]. Therefore, our process has great potential to produce scalable aBN layers across a large range of thicknesses.

Figure R3.1: Raman spectra of (a) 10 nm aBN deposited at 225 °C on SiO₂ (blue) and bare SiO₂ substrate reference (black) and (b) 30 nm aBN deposited at 200 °C on Si.

3. To clarify the thickness of the aBN film, the authors have to show the height profiles of their aBN films as a function of deposition temperature using atomic force microscopy.

We thank the reviewer for the suggestion. We note that while atomic force microscopy (AFM) is generally used for the thickness of the layered hBN phase, it is less ideal for the aBN film. In the case of CVD-grown hBN film or exfoliated hBN flakes, AFM scan can be performed at the flake-substrate boundary to determine the hBN thickness. However, since the phase of the ALD-BN is amorphous, and the film is uniformly coated and extends to the edge of the substrate, no film-substrate boundary is present for the AFM thickness characterization. We have provided the thickness of aBN as a function of deposition temperature as determined by spectroscopic ellipsometry (SE) in **Supplementary Figure 3a**.

In addition, we prepared cross-sectional samples and used HAADF-STEM and SEM to characterize their height profiles and crystallinity. A set of cross-sectional images for ~ 2, 8, and 11 nm ALD-BN films is provided (**Figure R3.2**). While doing AFM on such uniformly coated films usually requires us to scratch

the surface off to make a measurable height, we believe that TEM or SEM and SE will be better alternative for its thickness characterization.

Figure R3.2: HAADF and SEM images from 2, 8, and 11 nm thick aBN films for thickness and crystallinity confirmation.

4. In the aBN/MoS₂ quantum well stack structure, what is the difference between hybrid ALD (65 °C and 250 °C) and 250 °C ALD for a device?

ALD of aBN at a deposition temperature of 250 °C results in a discontinuous morphology of the aBN film on MoS₂, as illustrated in **Supplementary Figure 5a**. Hybrid ALD (deposition temperature at 65 °C, then at 250 °C) allows for the continuous deposition of aBN on MoS₂ (**Supplementary Figure 5c**), thus making this approach more ideal for fabricating the aBN/MoS₂ quantum well stack structures. A continuous aBN morphology formed via the hybrid ALD process ensures the uniform separation of MoS₂ layers for maintaining electronic isolation across a large area.

5. In Figure 3e, the EDS map of N is unclear to readers. It should be revised to a clear image.

We thank the reviewer for the suggestion. We modified **Main Text Figure 3e** to show a clearer representation of the EDS map of N.

REVIEWERS' COMMENTS

Reviewer #1 (Remarks to the Author):

The authors have addressed my comments in detail, and revised the manuscript accordingly. Now, I recommend its publication.

Reviewer #2 (Remarks to the Author):

1.It would be advisable to remove the claim of "high performance" due to the observed absence of a high on-state current alongside rapid switching capabilities and low power consumption, which are critical for assessing intrinsic gate delay and energy-delay product.

2.Could the authors establish a correlation between the interface trap density and the defect density in amorphous BN (aBN)? Given that the defect density in MOCVD-grown MoS₂ is widely acknowledged to be on the order of 10^{12} - 10^{13} cm⁻², which invariably influences the interface trap properties, it would be beneficial to elucidate how these effects can be distinguished from one another.

Reviewer #3 (Remarks to the Author):

In the revised manuscript, additional experiments have been done to answer the reviewer's concerns. Now the authors have provided more data sets and further information on the relationship between growth temperature and aBN in the point-by-point reply. However, I strongly recommend revising the main text with the answers to question 1 (The reason for the higher dielectric constant compared with other aBN reports) to make this manuscript clear.

Furthermore, their HAADF and SEM images do not display clear thickness in Figure R3.2. In particular, SEM image makes it confusing. The authors completed the transfer process of their aBN film, as shown in Figure 3a. They can find film-substrate boundary areas to measure height profiles by AFM. Moreover, they should add the spectroscopic ellipsometry data in Supplementary Figure 3 to determine the ellipsometry thickness.

I would recommend acceptance after the authors tidy up the above in text.

RESPONSE TO REVIEWERS' COMMENTS

We thank all three referees for reviewing our paper. Here's our response to their remarks.

Reviewer #1 (Remarks to the Author):

The authors have addressed my comments in detail, and revised the manuscript accordingly. Now, I recommend its publication.

ANS: We thank the reviewer for the review and recommendation.

Reviewer #2 (Remarks to the Author):

Q1. It would be advisable to remove the claim of "high performance" due to the observed absence of a high on-state current alongside rapid switching capabilities and low power consumption, which are critical for assessing intrinsic gate delay and energy-delay product.

ANS: We thank the reviewer for this suggestion. Although we do not demonstrate the capabilities in rapid-switching and power consumption, we have shown significant improvement in the off-state behavior of the device via the development of an aBN dielectric stack, which is the primary focus of this work. Our title intends to convey the development of improved dielectric gate stacks that is ultimately also crucial to realize high-performance 2D electronics.

Q2. Could the authors establish a correlation between the interface trap density and the defect density in amorphous BN (aBN)? Given that the defect density in MOCVD-grown MoS₂ is widely acknowledged to be on the order of 10¹²-10¹³ cm⁻², which invariably influences the interface trap properties, it would be beneficial to elucidate how these effects can be distinguished from one another.

ANS: We thank the reviewer for the suggestion of a future study and will investigate the defect density of aBN using techniques, such as STM/STS spectroscopy, and carry out theory study to understand what defects can appear in aBN for furthering the development of aBN dielectrics for 2D electronics.

Reviewer #3 (Remarks to the Author):

In the revised manuscript, additional experiments have been done to answer the reviewer's concerns. Now the authors have provided more data sets and further information on the relationship between growth temperature and aBN in the point-by-point reply. However, I strongly recommend revising the main text with the answers to question 1 (The reason for the higher dielectric constant compared with other aBN reports) to make this manuscript clear.

ANS: We thank the reviewer for this suggestion. We have added the supporting discussion on aBN dielectric constant on page 10 of the **Main Text**.

Furthermore, their HAADF and SEM images do not display clear thickness in Figure R3.2. In particular, SEM image makes it confusing. The authors completed the transfer process of their aBN film, as shown in Figure 3a. They can find film-substrate boundary areas to measure height profiles by AFM. Moreover, they should add the spectroscopic ellipsometry data in Supplementary Figure 3 to determine the ellipsometry thickness.

ANS: To provide clarity, we have added the spectroscopic ellipsometry data in **Supplementary Figure 3**.

I would recommend acceptance after the authors tidy up the above in text.

ANS: We thank the reviewer for the review and recommendation.